# Not All Weight Loss Is Equal: Divergent Patterns and Prognostic Roles in Head and Neck Cancer Versus High-Grade B-Cell Lymphoma

**DOI:** 10.3390/nu17152530

**Published:** 2025-07-31

**Authors:** Judith Büntzel, Gina Westhofen, Wilken Harms, Markus Maulhardt, Alexander Casimir Angleitner, Jens Büntzel

**Affiliations:** 1Department of Hematology and Medical Oncology, University Hospital, Robert-Koch-Straße 40, 37075 Göttingen, Germanymarkus.maulhardt@med.uni-goettingen.de (M.M.);; 2Department of Otolaryngology, Head Neck Surgery, Südharz Hospital, 99734 Nordhausen, Germany; jens.buentzel@shk-ndh.de

**Keywords:** cancer cachexia, weight change, head and neck cancer, high-grade B-cell lymphoma, cancer survivorship, inflammation markers

## Abstract

**Background:** Malnutrition and unintended weight loss are frequent in cancer patients and linked to poorer outcomes. However, data on long-term weight trajectories, particularly comparing different cancer entities, remain limited. **Methods:** In this retrospective, multicenter study, we analyzed 145 patients diagnosed with either head and neck cancer (HNC; *n* = 48) or high-grade B-cell lymphoma (HGBCL; *n* = 97). Body weight, C-reactive protein (CrP), albumin, and modified Glasgow Prognostic Score (mGPS) were assessed at diagnosis and at 3, 6, 9, and 12 months. Clinically relevant weight loss was defined as >5% from baseline. Survival analyses were performed for HGBCL patients. **Results:** Weight loss was common in both cohorts, affecting 32.2% at 3 months and persisting in 42.3% at 12 months. Nearly half of HNC patients had sustained >5% weight loss at one year, whereas HGBCL patients were more likely to regain weight, with significantly higher rates of weight gain at 6 and 12 months (*p* = 0.04 and *p* = 0.02). At baseline, HGBCL patients showed elevated CrP and lower albumin compared to HNC (both *p* < 0.001). Weight loss at 6 months was significantly associated with reduced overall survival in HGBCL (*p* < 0.01). Both Δweight at 6 months and mGPS emerged as useful prognostic indicators. **Conclusions:** This study reveals distinct patterns of weight change and systemic inflammation between HNC and HGBCL patients during the first year after diagnosis. These findings highlight the need for entity-specific nutritional monitoring and tailored supportive care strategies extending into survivorship. Prospective studies integrating body composition analyses are warranted to better guide long-term management.

## 1. Introduction

Malnutrition is a common occurrence in cancer patients, with up to 85% of individuals experiencing clinically relevant weight loss during treatment, depending on the cancer entity [1,2]. In addition to tumor type, the modality of cancer treatment also significantly influences weight trajectories throughout therapy [3]. Overall, malnutrition and weight loss are associated with worse prognosis, reduced quality of life, and an increased incidence of treatment-related toxicities [4,5,6,7,8].

With a growing number of cancer survivors, nutritional management has become a critical component of survivorship care, necessitating a multidisciplinary approach [9,10]. However, most current clinical guidelines—including the ESPEN clinical nutrition guideline for cancer patients [6,11] and the American Cancer Society’s nutrition and physical activity guideline for cancer survivors [12]—provide limited guidance on the long-term monitoring and management of weight and nutritional status beyond the active treatment phase. Tools such as the modified Prognostic Glasgow Score (mPGS) using albumin and C-reactive protein (CrP) apply nutrition and inflammation related markers for estimating prognosis in cancer patients [13,14]; however, this score is not a dynamic parameter suitable for follow-up.

Indeed, the majority of existing studies concentrate on weight changes occurring during active cancer treatment [15,16,17,18,19]. Particularly in lymphoma patients, clinical attention often centers on weight loss as part of the B-symptoms [18]. As yet, systematic follow-up data after completion of therapy remain scarce [20] and studies addressing weight trajectories in the survivorship phase predominantly target weight gain in breast cancer survivors [21,22].

Currently, the field underestimates the influence of the distinct pathophysiological mechanisms inherent to different cancer entities. For example, high-grade B-cell lymphoma (HGBCL) represents a systemic disease largely driven by inflammation, with immunochemotherapy being the standard of care [23,24]. In contrast, head and neck cancer (HNC) serves as a prime example of a malignancy where both the tumor itself and local treatments (e.g., surgery, radiotherapy) can directly impair nutritional intake due to local anatomical and functional disruption [25]. These two entities exemplify how differing disease characteristics may differentially affect nutritional status and metabolism. It is therefore reasonable to hypothesize that the course of body weight changes following diagnosis and treatment would also vary between cancer types.

Based on this hypothesis, the present study aimed to evaluate entity-specific patterns of weight loss and subsequent weight recovery in patients with HGBCL and HNC during the first twelve months after diagnosis. The findings are intended to inform and support the development of tailored survivorship care programs that address the nutritional needs of cancer patients in a differentiated manner.

## 2. Materials and Methods

As the aim of this study was to compare weight trajectories in two exemplary cancer entities (cancer as locoregional impairment vs. cancer as systemic disease), this retrospective, multicenter study included 48 patients diagnosed with HNC (treated between January 2020 and December 2022) and 97 patients diagnosed with HGBCL (between January 2021 and December 2022). Follow-up data were available until August 2023. Inclusion criteria were as follows: diagnosis of HNC or HGBCL, age ≥ 18 years, and availability of body weight measurements at a minimum of three different time points. The numbers of cases screened and included are depicted in Appendix A. Clinical data, including follow-up, survival, treatment modalities, laboratory parameters (CrP, albumin), and body weight, were retrospectively retrieved from electronic medical records. We further checked whether treating physicians used nutritional interventions during and after cancer treatment of patients. The study was conducted in accordance with the Strengthening the Reporting of Observational Studies in Epidemiology (STROBE) guidelines [26]; the corresponding checklist is provided in Appendix B. The study protocol was approved by the Ethics Committee of the University Medical Center Göttingen (approval number 12/6/24, approval date: 10 June 2024).

Relevant weight loss was defined as a decrease in body weight of >5%, in accordance with the international consensus criteria for cancer cachexia [27] and the Common Terminology Criteria for Adverse Events, Version 5.0 (CTCAE v5.0; [28]). We defined weight gain as an increase in body weight >5%. Patients with a decrease or increase in weight ≤5% were defined as patients with stable weight.

Body weight was assessed at five predefined time points: t_0_: at diagnosis, t_1_: 3 months after diagnosis, t_2_: 6 months after diagnosis, t_3_: 9 months after diagnosis, t_4_: 12 months after diagnosis. The rate of weight change (Δweight) was calculated as follows: Δweight = weight at t_x_ − weight at t_0_. Albumin and CrP levels were used to calculate the mPGS, a validated prognostic tool for cancer patients. The score is calculated as follows: albumin > 35 g/L and CrP < 10 mg/dL—0 points; albumin > 35 g/L, CrP > 10 mg/dL—1 point; albumin < 35 g/L, CrP > 10 mg/dL—2 points [13,14].

### Statistical Analysis and Data Handling

Data extraction and curation were performed using Microsoft Excel (Microsoft Excel 2021; Microsoft Corporation, Redmond, WA, USA). Descriptive statistics (mean, standard deviation, median, range, frequencies) were used to summarize demographic and clinical characteristics. Due to the retrospective nature of the study, missing data were present, particularly regarding body weight at individual follow-up time points. Analyses were conducted using the available data, and sample sizes were adjusted accordingly for each comparison. Statistical methods were chosen to account for non-normal distributions and potential effects of missing data. No imputation of missing values was performed.

Comparisons of categorical variables were conducted using Fisher’s exact test. The Mann–Whitney U test and Kruskal–Wallis test were applied for comparisons of non-parametric continuous variables, as appropriate. Survival analyses were performed using Kaplan–Meier estimators, with differences between groups assessed by the log-rank (Mantel–Cox) test. Kaplan–Meier curves were generated to visualize survival probabilities. The predictive value of body weight change (Δweight) for overall survival was evaluated using receiver operating characteristic (ROC) curve analysis. All statistical analyses were conducted using GraphPad Prism software (version 9.3.0; GraphPad Software, Boston, MA, USA). A two-tailed *p*-value < 0.05 was considered statistically significant; *p*-values between 0.05 and 0.15 were interpreted as a trend. The manuscript was reviewed and revised for language, grammar, style, and formatting using ChatGPT-4o, the June 2025 version of OpenAI’s large language model (OpenAI, L.L.C., San Francisco, CA, USA).

## 3. Results

### 3.1. Clinical Characteristics

A total of 145 patients were included in the analysis, comprising 48 patients with HNC and 97 patients with HGBCL. Age and body mass index (BMI) at baseline (time of cancer diagnosis) did not differ significantly between the two cohorts (Table 1). Among HNC patients, 14/48 underwent surgery alone, 11/48 received radio(chemo)therapy, and 23/48 were treated with surgery followed by adjuvant radio(chemo)therapy. HGBCL patients received systemic therapy according to national guidelines [24]. Within the HGBCL cohort, 11/97 received chimeric-antigen-receptor-T cell therapy and 19/97 underwent high-dose chemotherapy followed by autologous stem cell transplantation after failure of first-line treatment.

### 3.2. Weight Loss Is a Common Event in Cancer Patients

Weight loss is a frequent occurrence among cancer patients, with weight loss rates remaining relatively stable over the course of one year following diagnosis (Figure 1). At time-point t_1_, 32.20% of patients had experienced weight loss, and 42.31% still exhibited reduced weight compared to baseline (t_0_) one year later (t_4_). No significant differences in weight loss rates were observed between HNC and HGBCL patients. In both cohorts, the proportion of patients maintaining stable weight declined over time (HNC: t_1_ = 64.44% vs. t_4_ = 47.73%; HGBCL: t_1_ = 64.36% vs. t_4_ = 38.24%). Notably, HGBCL patients demonstrated a greater ability to regain weight over time, with significantly higher rates of weight gain observed at t_2_ (HNC: 0.00%; HGBCL: 13.33%; *p* = 0.04) and at t_4_ (HNC: 4.55%; HGBCL: 26.47%; *p* = 0.02, Table 1). For changes in Δweight over time refer to Appendix A.

Nutritional interventions were administered in 19.29% (27/140) of patients, with the vast majority belonging to the HNC cohort (26/27). Most interventions were initiated at baseline (t_0_: 27/140) and were less frequently applied at later time-points (t_1_: 12/140). Detailed information on the type of nutritional intervention was available for 26 patients: 9 received percutaneous endoscopic gastrostomy (PEG) feeding, 9 received intravenous nutrition, and 8 were supplemented with high-energy oral nutrition. Nutritional intervention at t_0_ did not significantly impact the rate of weight loss at t4 in HNC patients (*p* = 1.00).

### 3.3. Weight Loss Has No Influence on Recurrent Disease in HNC or HGBCL Patients, but on Survival

Overall, recurrent disease was documented in 10.42% (15/144) of patients. Weight loss between t_1_ and t_4_ was not significantly associated with recurrence in the overall study population. Subgroup analyses of HNC and HGBCL cohorts similarly revealed no increased risk of recurrence in patients experiencing weight loss.

Survival data were available for HGBCL patients. Those who exhibited weight loss at t_1_ (*p* = 0.052) and t_3_ (*p* = 0.073) showed a trend toward shorter overall survival compared to patients with stable weight or weight gain. Notably, patients with weight loss at t_2_ had significantly reduced overall survival (*p* < 0.01) (Figure 2).

While Δ(weight_t1−t0_) did not differ significantly between surviving and deceased HGBCL patients, significant differences were observed for Δ(weight_t2−t0_) and Δ(weight_t3−t0_) (Figure 3a). This effect was not seen at t_4_. Δ(weight_t2−t0_) emerged as a useful clinical marker for survival prediction, as demonstrated by ROC curve analysis (Figure 3b).

### 3.4. CrP and Albumin as Indicators for Systemic Inflammation in HGBCL Patients

CrP and albumin levels were compared between HNC and HGBCL patients. At baseline, HGBCL patients exhibited significantly higher CrP levels and lower albumin concentrations than HNC patients. Following initiation of systemic therapy, CrP levels in HGBCL patients became comparable to those in HNC patients, supporting the interpretation of elevated CrP as a marker of systemic inflammation. Nevertheless, albumin levels remained significantly lower in HGBCL patients throughout the first six months and continued to be lower even one year post-diagnosis (Table 2, Figure 4a,b).

The mGPS differed significantly between cohorts only at baseline, with a higher median score in HGBCL patients (median mGPS(HGBCL) = 1 [0–2] vs. mGPS(HNC) = 0 [0–2]; *p* < 0.001). Since survival data were available only for the HGBCL cohort, we assessed whether mGPS could predict poor outcomes. An mGPS of 2 was significantly associated with mortality in HGBCL patients at t_0_ (*p* = 0.031), t_1_ (*p* = 0.040), and t_2_ (*p* = 0.010) (Fisher’s exact test).

## 4. Discussion

In our retrospective study, we observed clinically relevant weight loss in both HNC and HGBCL patients over the first year following diagnosis. HGBCL patients demonstrated a more favorable trajectory, with many regaining weight, whereas nearly half of the HNC patients failed to return to their baseline weight. Both patterns—persistent weight loss among HNC survivors [29] and weight gain following HGBCL treatment—have been previously reported [20,30].

Persistent weight loss in HNC patients is well documented. Studies report rates of post-treatment weight loss reaching nearly 95% in this group [29,31,32]. The discrepancy between the 95% reported by Jin et al. and our lower proportion can likely be attributed to our stricter definition of weight loss, categorizing only patients with a significant reduction of >5% as “weight loss” cases.

At first glance, post-treatment weight gain in HGBCL patients appears more favorable than the prolonged weight loss observed in HNC patients. However, prior studies focusing on body composition have shown that lymphoma patients frequently lose muscle mass during treatment [20,33] while simultaneously gaining visceral fat. Notably, significant weight gain (>5%) has been linked to increases in visceral adiposity in HGBCL patients [20]. Moreover, weight gain itself has been discussed as an unfavorable prognostic marker. In a study on female non-Hodgkin lymphoma patients, those with weight gain after treatment had significantly shorter overall survival compared to patients maintaining stable weight [30].

We intentionally chose to compare these two cancer entities because weight loss is prevalent in both groups but driven by fundamentally different mechanisms. This distinction was also reflected in the inflammatory profiles of our cohorts. At diagnosis, HGBCL patients exhibited significantly higher CrP levels and lower albumin concentrations than HNC patients.

Inflammation plays a crucial role in HGBCL, shaping the tumor microenvironment and influencing treatment response [23]. Tumor-induced inflammation is a known driver of fat and muscle catabolism, ultimately leading to malnutrition and cancer cachexia [34]. Effective treatment of HGBCL often resolves this pro-inflammatory state, as evidenced by the decline in mean CrP and increase in albumin levels over time in our HGBCL cohort. By contrast, HNC patients primarily undergo local treatments such as surgery or radio(chemo)therapy aimed at tumor eradication [35]. Consequently, they face a different spectrum of side effects—including dysgeusia, xerostomia, dysphagia, and odynophagia—which are closely linked to persistent post-treatment weight loss [25,29].

These distinct patterns of weight change between HNC and HGBCL already highlight the need for supportive care strategies tailored to the specific challenges of each cancer type. Nutritional interventions are critical during treatment in patients with malnutrition, and routine screening is advocated by international guidelines [6,11]. However, our findings emphasize that monitoring weight trajectories should extend well into cancer survivorship. Both weight loss and weight gain have been shown to impact long-term survival in lymphoma patients [30]. The detrimental effects of weight gain may be partly explained by adverse changes in body composition, such as muscle loss accompanied by increased fat mass [30], which elevate the risk of metabolic syndrome and cardiovascular disease [36,37,38]. Considering that nearly half of HNC patients in our study exhibited > 5% weight loss compared to baseline even twelve months after diagnosis, it is essential to enhance supportive care efforts addressing long-term sequelae of cancer treatment. Preventing ongoing weight loss has itself been shown to positively influence both the physical and mental health of cancer survivors [39,40].

In line with previous studies [41,42,43], we were able to demonstrate that weight loss serves as a predictor of poorer overall survival. As noted, cancer-associated inflammation is closely tied to body mass loss [34]. Various prognostic scores incorporating inflammatory markers have been proposed to predict overall survival in HGBCL [23], yet none have been adopted into routine clinical practice. Our findings show that (1) Δweight six months after diagnosis effectively predicts patient survival, and (2) mGPS is a simple and well-established tool also suitable for HGBCL.

To our knowledge, this is the first study comparing real-world data on both HNC and HGBCL patients, providing insight into their distinct weight trajectories. Although weight data were not available for every patient at each time point throughout the one-year observation period, we still identified significant differences between the two cohorts, including during post-treatment recovery. By incorporating inflammatory biomarkers, we further established the connection between systemic inflammation and weight loss.

While the use of real-world data enhances the external validity of our results, several limitations must be acknowledged. The retrospective design of our study inherently introduces risks of selection bias, incomplete data capture, and limitations in causal inference. Inclusion was restricted to patients with weight measurements at multiple time points, potentially over-representing individuals with better follow-up adherence and survival outcomes. Due to the limited number of patients and the exploratory character of our study, we chose a simple design forgoing subgroup analysis that included prognostic features. Additionally, data were not consistently available at all planned time points, and particularly small sample sizes at t3 and t4 may have affected statistical power and the robustness of conclusions. Moreover, the absence of body composition assessments leaves unresolved questions about the qualitative nature of observed weight changes. Future research should consider prospective designs incorporating both anthropometric data and body composition analyses (e.g., via computer tomography scans or bio-impedance analysis).

## 5. Conclusions

This study highlights distinct patterns of weight change and systemic inflammation in patients with head and neck cancer (HNC) and high-grade B-cell lymphoma (HGBCL) over the first year following diagnosis. While weight loss was prevalent in both groups, HGBCL patients showed a greater capacity for regaining weight, whereas nearly half of HNC patients continued to exhibit significant weight loss one year post-diagnosis. Importantly, we identified Δweight at six months and the mGPS as useful tools for predicting survival outcomes in HGBCL. These findings underscore the necessity of entity-specific nutritional monitoring and supportive care strategies that extend well beyond active treatment, aiming to address the unique metabolic and inflammatory challenges associated with different cancer types. Prospective studies incorporating detailed assessments of body composition are warranted to further optimize long-term survivorship care.

## Figures and Tables

**Figure 1 nutrients-17-02530-f001:**
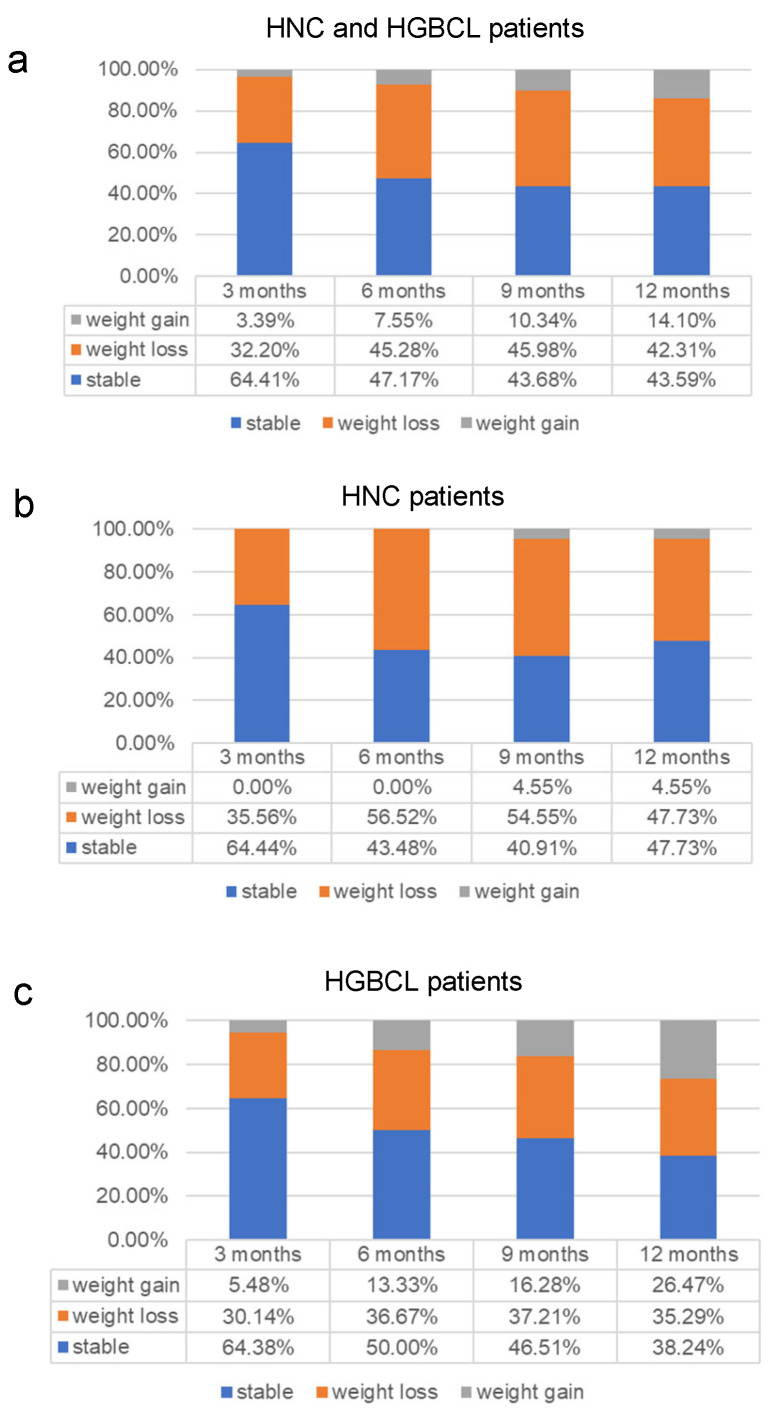
Weight trajectory of patients with head and neck cancer (HNC) or high-grade B-cell lymphoma (HGBCL); (**a**) weight loss over time is a common event in cancer patients. Subgroup analysis of HNC (**b**) and HGBCL (**c**).

**Figure 2 nutrients-17-02530-f002:**
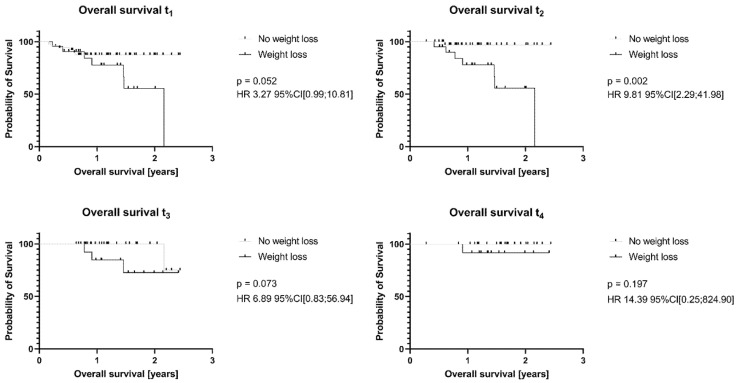
Weight loss at t_2_ is signifcantly associated with reduced survival in patients with high grad B-cell lymphoma. Kaplan-Meier Curve, log-rank (Mantel–Cox) test. HR—hazard ratio; CI—confidence intervall.

**Figure 3 nutrients-17-02530-f003:**
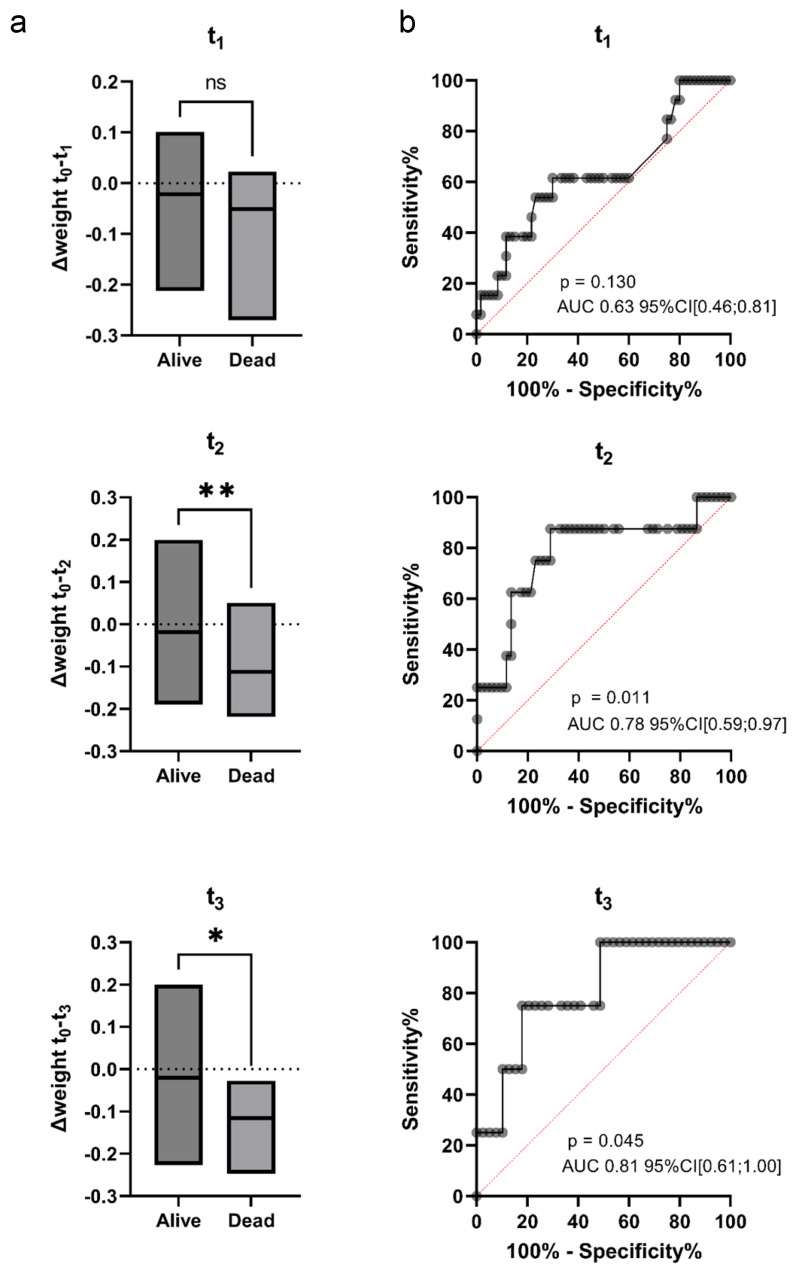
Δweight is a predictor of outcome in high-grade B-cell lymphoma (HGBCL) patients. (**a**) Plots comparing Δweight of survivors and deceased patients at t_1_–t_3_. (**b**) Receiver Operator Curve (ROC) analysis of the corresponding Δweight time-points. *—*p* < 0.05, **—*p* < 0.01.

**Figure 4 nutrients-17-02530-f004:**
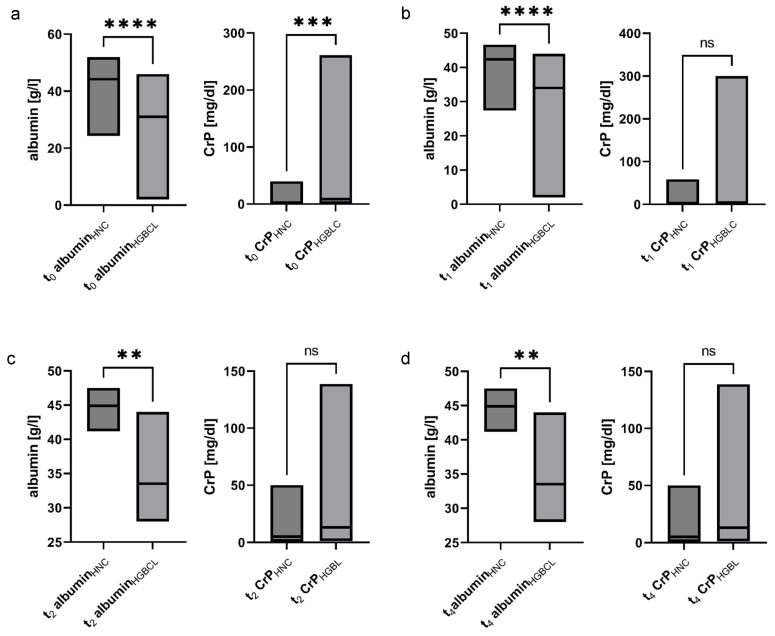
Head and neck cancer (HNC) and high-grade B-cell lymphoma (HGBCL) show significantly lower albumin and C-reactive protein (CrP) levels at t_0_ (**a**). Albumin levels remain significantly lower in HGBCL patients over the course of three months (**b**), six months (**c**) and twelve months (**d**). **—*p* ≤ 0.01, ***—*p* ≤ 0.001, ****—*p* ≤ 0.0001.

**Table 1 nutrients-17-02530-t001:** Clinical characteristics and weight development. HNC—head and neck cancer, HGBCL—high-grade B-cell lymphoma, SD—standard deviation. HNC and HGBCL patients were compared to detect differences between cohorts. Bold was used for subheadings and significant *p*-values.

	HNC + HGBCL	HNC	HGBCL	*p*-Value
**N [patients]**	145	48	97	
**Sex**				
**Male**	101	41	60	
**Female**	44	7	37	
**Age [years] +/− SD**	63.08 +/− 12.77	61.89 +/− 9.73	63.67 +/− 13.99	0.43
**Body-Mass-Index [kg/m^2^] +/− SD**	25.78 +/− 7.31	26.42 +/− 5.87	27.23 +/− 2.18	0.46
**T_1_ (3 months)**				
Stable weight, N [patients]	76/118 (64.41%)	29/45 (64.44%)	47/73 (64.38%)	
Weight loss, N [patients]	38/118 (32.20%)	16/45 (35.56%)	22/73 (30.14%)	0.69
Weight gain, N [patients]	4/118 (3.39%)	0/45 (0.00%)	4/73 (5.48%)	0.29
**T_2_ (6 months)**				
Stable weight, N [patients]	50/106 (47.17%)	20/46 (43.48%)	30/60 (50.00%)	
Weight loss, N [patients]	48/106 (45.28%)	26/46 (56.52%)	22/60 (36.67%)	0.22
Weight gain, N [patients]	8/106 (7.55%)	0/46 (0.00%)	8/60 (13.33%)	**0.04**
**T_3_ (9 months)**				
Stable weight, N [patients]	38/87 (43.68%)	18/44 (40.91%)	20/43 (46.51%)	
Weight loss, N [patients]	40/87 (45.98%)	24/44 (54.55%)	16/43 (37.21%)	0.26
Weight gain, N [patients]	9/87 (10.34%)	2/44 (4.55%)	7/43 (16.28%)	0.27
**T_4_ (12 months)**				
Stable weight, N [patients]	34/78 (43.59%)	21/44 (47.73%)	13/34 (38.24%)	
Weight loss, N [patients]	33/78 (42.31%)	21/44 (47.73%)	12/34 (35.29%)	1.00
Weight gain, N [patients]	11/78 (14.10%)	2/44 (4.55%)	9/34 (26.47%)	**0.02**
**B-symptoms**				
B symptoms, N [patients]	33/130	0/48	33/81	**0.0001**
Nutritional intervention t_0_ N [patients]	27/140	26/48	1/92	**0.0001**
Nutritional intervention t_1_ N [patients]	12/140	11/48	1/92	**0.0001**

**Table 2 nutrients-17-02530-t002:** Changes in C-reactive protein and albumin over time. HNC—head and neck cancer, HGBCL—high-grade B-cell lymphoma, CrP—C-reactive protein. Bold was used for subheadings and significant *p*-values.

	HNC + HGBCL	HNC	HGBCL	*p*-Value
**t_0_ (diagnosis of cancer)**				
CrP [mg/dL]	26.65 ± 51.17	5.64 ± 8.38	37.03 ± 59.60	**<0.001**
albumin [mg/L]	35.03 ± 10.64	44.15 ± 4.36	30.01 ± 9.70	**<0.001**
**t_1_ (3 months)**				
CrP [mg/dL]	18.92 ± 33.72	16.51 ± 27.29	20.13 ± 36.45	0.99
albumin [mg/L]	33.39 ± 11.47	41.48 ± 4.92	30.08 ± 11.74	**<0.001**
**t_2_ (6 months)**				
CrP [mg/dL]	24.84 ± 62.45	8.87 ± 14.42	32.13 ± 73.59	0.18
albumin [mg/L]	34.47 ± 10.56	41.16 ± 5.67	31.31 ± 10.85	**<0.001**
**t_3_ (9 months)**				
CrP [mg/dL]	10.48 ± 16.81	15.05 ± 12.18	9.28 ± 17.63	0.12
albumin [mg/L]	29.73 ± 13.01	38.78 ±6.79	27.14 ± 13.21	0.75
**t_4_ (12 months)**				
CrP [mg/dL]	24.75 ± 37.67	14.68 ± 18.02	31.72 ± 45.36	0.35
albumin [mg/L]	37.46 ± 10.51	44.65 ± 2.34	31.70 ± 10.95	**0.003**

## Data Availability

Due to patient sensitive data and study protocol as well as require-ments of the Institutional Review Board of the University Medical Center, the data presented in this study are available on request from the corresponding author.

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
