# Peer review of "Not All Weight Loss Is Equal: Divergent Patterns and Prognostic Roles in Head and Neck Cancer Versus High-Grade B-Cell Lymphoma"

_nutrients, 2025, doi:10.3390/nu17152530_

Round 1
Reviewer 1 Report
Comments and Suggestions for Authors
I thank the authors for their manuscript, which I have read very carefully. The subject covered is of great interest to the scientific community. The introductory part has been written in a precise, brief and focused manner to introduce the research question. The part on materials and methods describes in a very brief way the methodology of the study, perhaps in this part it would have been better to write in a more extensive and thorough manner the methodology and materials used for the conduct of the study. In the results section, the results are reported both as text and as data representation in tables and figures. I find Table 1 uninteresting. To improve this table, I propose that the authors indicate both the number (absolute term) and the percentage (%, relative term). The BMI (average/SD) is shown in the table, followed by the weight (stable, decrease, increase). In this section of the table I suggest to report for T1,T2,T3,T4: a) number of cases, b) (percentage of cases), c) average weight, d) SD of mean, e) delta of mean weight, f) SD of mean weight. Reporting only the % of cases (stable weight, weight loss, weight gain) I do not think is such a striking figure, it seems to me more important that it is also reported of how many kg on average/ SD are varied cases. The part of the discussion seems to be adequate for the purpose of the manuscript, as well as the conclusions. The limitations of the study have been reported. Appendix A, for example, I would remove it from the manuscript.
Author Response
For Authors
I thank the authors for their manuscript, which I have read very carefully. The subject covered is of great interest to the scientific community. The introductory part has been written in a precise, brief and focused manner to introduce the research question.
The part on materials and methods describes in a very brief way the methodology of the study, perhaps in this part it would have been better to write in a more extensive and thorough manner the methodology and materials used for the conduct of the study.
Answer 1:
We read our method section carefully, and added the following passage to clarify the definition of stable weight and weight gain: “We defined weight gain as an increase in body weight > 5%. Patients with a decrease or increase in weight ≤ 5% were defined as patients with stable weight.”
The remaining parts were checked carefully, as the design of our study is pretty simple, we were not sure where you would like to have more details. If you have further comments/ ideas on this matter, we are happy to adjust the section accordingly.
In the results section, the results are reported both as text and as data representation in tables and figures. I find Table 1 uninteresting. To improve this table, I propose that the authors indicate both the number (absolute term) and the percentage (%, relative term).
The BMI (average/SD) is shown in the table, followed by the weight (stable, decrease, increase). In this section of the table I suggest to report for T1,T2,T3,T4: a) number of cases, b) (percentage of cases), c) average weight, d) SD of mean, e) delta of mean weight, f) SD of mean weight.
Answer 2:
We added the absolute numbers of patients for all four time points to Table 1.
BMI and absolute weight do not reflect changes overtime and do not reflect malnutrition, especially if we use the mean and SD – therefore we chose not to add these parameters after t0. However, we added delta(weight) change to Supplementary Table 1, which is now part of Appendix A.
Reporting only the % of cases (stable weight, weight loss, weight gain) I do not think is such a striking figure, it seems to me more important that it is also reported of how many kg on average/ SD are varied cases.
Answer 3:
Reporting kg on average is not representive, as every patient has their own “baseline weight” at t0. A weight loss of 5 kg in a patient weighing 120 kg is not significant, while a weight loss of 5 kg has an impact on a patient with 50 kg at baseline. Therefore, we had to “normalize” the data using Δweight to compare patients and to reflect the dynamic each patient had over time.
The part of the discussion seems to be adequate for the purpose of the manuscript, as well as the conclusions. The limitations of the study have been reported.
Answer 4:
Thank you for you kind words.
Appendix A, for example, I would remove it from the manuscript.
Answer 5:
Appendix A with the STROBE list is a requirement of the journal, therefore we kept it.
Reviewer 2 Report
Comments and Suggestions for Authors
This study compares weight loss and overall survival between patients with HNC and HGBCL. Overall, the study is relatively simple in scope.
The quality of the introduction needs improvement. For example, is cancer-related weight loss and treatment-induced weight loss common? What are the severe consequences? What is the urgency and necessity of nutritional intervention?
The discussion on the value of nutritional intervention in cancer is also too weak and fails to emphasize its importance.
The introduction does not sufficiently explain why albumin and creatine phosphate are being measured. These are key indicators and central to your research focus.
Methods: Please clearly describe the grouping and intervention methods. These details are currently insufficient—I only understood your study design after reading the results section.
Figures: Ensure consistent font usage—please use either Times New Roman or Arial throughout. Also, change all fonts to black.
Please capitalize the first letter of each word in the figure legends and on all axis labels.
Lines 210–212: Please explain these findings. Why does nutritional intervention lead to these changes, especially in albumin concentration?
Line 236: It is necessary to cite relevant studies to support this point. Merely describing the findings is not sufficient.
Discussion: You should compare your findings with previous studies on cancer-related weight loss and explain how your results differ from past research.
What guidance or practical implications does your study offer for cancer patients?
Table 1: Why is the information so limited? Past medical history and other clinical data are also very important and should be included.
Please pay attention to abbreviations. Abbreviations must be defined at first use in the abstract, main text, and tables, and can then be used consistently throughout the manuscript.
Please update outdated references. Some of the references cited are more than five years old.
Author Response
Comment 1:
This study compares weight loss and overall survival between patients with HNC and HGBCL. Overall, the study is relatively simple in scope.
Answer 1:
Yes, the design is simple, however the implications are not.
We demonstrate, that weight loss is relevant after the end of treatment. This implies that patients have less resources to fall back (weight loss is often a combination of muscle and fat loss), in case of a second event (recurrent disease, other events that require hospitalization). Further, weight gain, especially in HGBCL patients has implications for follow-up – gain in weight is often a gain fat mass (we discussed the studies about body composition), which itself may lead to metabolic syndrome and is associated with cardiovascular events after surviving cancer. Our study demonstrates, that a simple parameter as the body weight is a good indicator how to manage patients in follow-up and to improve cancer survivorship. Weight loss ïƒ nutritional support, physical therapy to regain muscle mass +/- nutritional intervention; weight gain ïƒ weight management, physical activity.
Comment 2:
The quality of the introduction needs improvement. For example, is cancer-related weight loss and treatment-induced weight loss common? What are the severe consequences? What is the urgency and necessity of nutritional intervention?
Answer 2:
Our first paragraph of the Introduction (p 2 l37ff) clearly addresses the importance of the topic:
“Malnutrition is a common occurrence in cancer patients, with up to 85% of individu-als experiencing clinically relevant weight loss during treatment, depending on the cancer entity [1,2]. In addition to tumor type, the modality of cancer treatment also significantly influences weight trajectories throughout therapy [3]. Overall, malnutrition and weight loss are associated with worse prognosis, reduced quality of life, and an increased incidence of treatment-related toxicities [4–8].”
Comment 3:
Could you please clarify, which part of the paragraph should be extended? Malnutrtion is common, patients with malnutrition or cancer cachexia die faster compared to patients with stable weight and have more side effects.
The discussion on the value of nutritional intervention in cancer is also too weak and fails to emphasize its importance.
Answer 3:
The aim of our study was not to demonstrate the effect of a nutrtional intervention. It was to demonstrate that different pathophysiological patterns of malnutrition (localized cancer -->local impairment, local treatment --> local problems – head neck cancer and cancer as systemic disease --> systemic inflammation --> High grade b cell lymphoma) lead to different weight -trajectories and therefore have different implications for nutritional intervention, follow-up, support and cancer survivorship. This main idea is reflected by our discussion.
Comment 4:
The introduction does not sufficiently explain why albumin and creatine phosphate are being measured. These are key indicators and central to your research focus.
Answer 4:
Creatinine phosphate was not measured. We used albumin and C-reactive protein as both established parameters for calculating the modified Prognostic Glasgow Score. We added a short paragraph to the introduction, introducing the modified Prognostic Glasgow Score.
“Tools such as the modified Prognostic Glasgow Score (mPGS) using albumin and C-reactive protein (CrP) apply nutrition and inflammation related markers for estimating prognosis in cancer patients [26,27], however this score is not a dynamic parameter suita-ble for follow-up.”
Comment 5:
Methods: Please clearly describe the grouping and intervention methods. These details are currently insufficient—I only understood your study design after reading the results section.
Answer 5:
We added a short sentence to the first paragraph of the method section to clarify, that we are interested in comparing head neck cancer patients (cancer as locoreginal impairment) to high grade B cell lymphoma patients (Cancer as systemic disease), p 3, l77f.
Comment 6:
Figures: Ensure consistent font usage—please use either Times New Roman or Arial throughout. Also, change all fonts to black.
Answer 6:
We are restricted to the graphical programs we are using, all Figures except Figure 1 use the same font. Figure 1 does not use Times New Roman (it was Calibri), we changed it.
Comment 7:
Please capitalize the first letter of each word in the figure legends and on all axis labels.
Answer 7:
The first letter of the beginning word is capitalized in all figure legends except for the Δweight – however, Δ is a capital letter itself in the Greek alphabet, so we do not understand this first part of your comment. Capitalizing each word of the figure legend text is not required by the journal’s author guideline. Axis labels were chosen as required by the journal.
Comment 8:
Lines 210–212: Please explain these findings. Why does nutritional intervention lead to these changes, especially in albumin concentration?
Answer 8:
We do not investigate nutritional interventions at all. We discuss different patterns of disease which are reflected by different weight trajectories and different laboratory findings. In the case of albumin – lower albumin levels are in high grade b cell lymphoma patients are probably associated with the inflammatory, systemic disease of HGBCL, which we discuss p9 l 223ff
Comment 9:
Line 236: It is necessary to cite relevant studies to support this point. Merely describing the findings is not sufficient.
Answer 9:
We do not understand this comment, as we cite three different studies (reference 41-43) to support our claims. You find this paragraph due to editing/revising with the references now in line 246.
Comment 10:
Discussion: You should compare your findings with previous studies on cancer-related weight loss and explain how your results differ from past research.
Answer 10:
We do compare to previous studies. However, keep in mind, that the difference of our study is, that we looked at data at more time points and compared to different entities to address differences in pathophysiological mechanisms of malnutrition.
Please refer to: “Both weight loss and weight gain have been shown to impact long-term survival in lymphoma patients [29]. The detrimental effects of weight gain may be partly explained by adverse changes in body composition, such as muscle loss accompanied by increased fat mass [30], which elevate the risk of metabolic syndrome and cardiovascular disease [36–38]. Considering that nearly half of HNC patients in our study exhibited > 5% weight loss compared to baseline even twelve months after diagnosis, it is essential to enhance supportive care efforts addressing long-term sequelae of cancer treatment. Preventing ongoing weight loss itself has been shown to positively influence both the physical and mental health of cancer survivors [39,40]“
As how our research differs, please refer to: “To our knowledge, this is the first study comparing real-world data on both HNC and HGBCL patients, providing insight into their distinct weight trajectories.”
Comment 11:
What guidance or practical implications does your study offer for cancer patients?
Answer 11:
As we wrote in our conclusion: “These findings underscore the necessity of entity-specific nutritional monitoring and supportive care strategies that extend well beyond active treatment, aiming to address the unique metabolic and inflammatory challenges associated with different cancer types. Prospective studies incorporating detailed assessments of body composition are warranted to further optimize long-term survivorship care.”
-->Implication for clinical practice are 1) malnutrition is different depending on cancer entity – not one size fits all but, individual follow-up; 2) weight gain in HGBCL is a common event, previous studies imply worse survival in this group --> monitoring in cancer survivorship, nutritional support, cardiovascular risk; 3) weight loss in HNC is common, remains and patients with weight loss have lower quality of life according to previous finding --> here we require nutritional support to regain weight and a completely different approach compared to the lymphoma group
Comment 12:
Table 1: Why is the information so limited? Past medical history and other clinical data are also very important and should be included.
Answer 12:
It is a retrospective study, relevant data was included. Information on treatment modalities (“other clinical data) of the cancer patients is listed in the text (“Among HNC patients, 14/48 underwent surgery alone, 11/48 received radio(chemo)therapy, and 23/48 were treated with surgery followed by adjuvant radio(chemo)therapy. HGBCL patients received systemic therapy according to national guidelines [21]. Within the HGBCL cohort, 11/97 received CAR-T cell therapy and 19/97 underwent high-dose chemotherapy followed by autologous stem cell transplantation after failure of first-line treatment”)
Comment 13:
Please pay attention to abbreviations. Abbreviations must be defined at first use in the abstract, main text, and tables, and can then be used consistently throughout the manuscript.
Answer 13:
Abbreviations are defined after first use as it is required abstract and main text are separate text bodies, therefore abbreviations are introduced twice (abstract and main text). We revised the main text for consistency. Abbreviation are introduced to each table and each figure separately are required by the journal’s author guidelines, we checked again and added missing details.
Comment 14:
Please update outdated references. Some of the references cited are more than five years old.
Answer 14:
Yes, some references are older, however this does not make the studies we cite invalid. Data on weight trajectories is scarce, we included studies, that were addressing this problem in similar populations (e. g. the lymphoma studies – we did not find more recent studies, which underlines the importance of our study)
Reviewer 3 Report
Comments and Suggestions for Authors
This study analyzed weight loss/gain in high-grade b-cell lymphoma and head and neck cancer. The text is well written, it is easy to read and understand. However, not sure why authors want to mix carcinoma with lymphoma. In addition, survival data is only available in lymphoma. There are several clinicopathological variables that define HNC and HGBCL, with correlation to histological, clinical and prognostic features of the patients that are not evaluated. It is fine, but these are limitattions to be acknowledged.
Additional comments:
(1) Lines 38-43. How is malnutrition defined?
(2) What are the cases of malnutrition in cancer? Is the intake reduced, or the energy usage increased bc of neoplasia?
(3) Instead of referencing the guidelines of ESPEN and ACS, it is possible to show them as table in appendix?
(4) Line 55. Only in breast cancer? Could you please confirm?
(5) Line 59. Regarding "HGBCL) represents a systemic disease largely driven by inflammation". Please explain in detail the meaning of "inflammation" in your context. HGBCL is an hematological neoplasia within the spectrum of diffuse large b-cell lymphoma (intermediate with Burkitt/LBL).
(6) Line 68. I understand using HNC, but why HGBCL? HGBCL is quite a rare disease and not representative of the most common hematological neoplasia.
(7) Please introduce clinicopathological characteristics of both HNC adn HGBCL in the Introduction sections.
(8) Line 86. Please add the statement of Helsinki for human experimentation.
(9) Line 94. Could you please add the mPGS in the Appendix as well?
(10) Line 103. How many cases had missing data? What percentage of the data was missing?
(11) Line 110. Why not using COX as well for Aweight predictive value?
(12) Line 113. I understand that "trend" is not significant.
(13) Line 121-122. Please describe HGBCL treatment (high-dose chemotherapy).
(14) Figure 1 is in 3D. I would make it as 2D as the third dimension is not giving any additional informtion.(Of note, this comment is just a matter of style).
(15) In Figure 1, should time zero (at diagnosis) be added as well?
(16) In Table 1. I am not sure of the meaning of variable "intervention T0 and T1".
(17) Figure 2 shows OS at differen times. As I understandn, T2 is at 6 months. I may be easier to remember "T 6months" instead of "T2".
(18) OS analysis was only done in HGBCL cases. Not having HNC data is a weak point.
(19) Figure 3 shows the ROC analysis. However, difference of weight is also shown in Figure 2 with the K-M and log rank analysis, isn't it? Is not ROC analysis redundant?
(20) Line 168. C-reactive protein is a marker of inflammation. However, is albumin as well? Please confirm.
Author Response
Comment 1
This study analyzed weight loss/gain in high-grade b-cell lymphoma and head and neck cancer. The text is well written, it is easy to read and understand.
However, not sure why authors want to mix carcinoma with lymphoma.
Answer 1:
As we outlined in the introduction as well as the conclusions, we chose high gade b cell lymphoma and head neck cancer as exemplary diseases for a cancer as systemic disease (high grade b cell lymphoma) and cancer as locoregional disease (head neck cancer), postulating that this two constellations may result in different patterns of malnutrition or weight development and therefore have different implications for cancer survivorship.
We added a short section (p. 7 l 77f) to clarify: “As the aim of this study was to compare weight trajectories in two exemplary cancer entities (cancer as locoregional impairment vs. cancer as systemic disease) […]”.
Comment 2:
In addition, survival data is only available in lymphoma. There are several clinicopathological variables that define HNC and HGBCL, with correlation to histological, clinical and prognostic features of the patients that are not evaluated. It is fine, but these are limitattions to be acknowledged.
Answer 2:
We added a small section to the limitations (“Due to limited patient number and the exploratory character of our study, we chose a simple design forgoing subgroup analysis that included prognostic features”). We also like to point out, that while histologically high grade b cell lymphoma and head neck cancer can be further characterized by mutations (HGBCL) or HPV status or by cancer side (larynx, hypopharynx, oropharynx ect.), studies tend to combine cases under the monikers/groups we used and treatment is usually similar, so that we postulated comparability between all patients of the HGBCL and HNC group.
Additional comments:
Comment 3:
(1) Lines 38-43. How is malnutrition defined?
Answer 3:
As we were concentrating on weight trajectories, we used relevant weight loss (> 5% in 3 months) as surrogate marker for malnutrition (refer. While malnutrition is defined by a quantitative or qualitative deficiency of nutrition (ESPEN Guideline), we only had the weight available for analysis, which itself reflects clinical routine, where usually weight and weight loss is assessed by professionals.
The method section has a paragraph reflecting this:
“Relevant weight loss was defined as a decrease in body weight of > 5%, in accordance with the international consensus criteria for cancer cachexia [24] and the Common Ter-minology Criteria for Adverse Events, Version 5.0 (CTCAE v5.0; [25]). We defined weight gain as an increase in body weight > 5%. Patients with a decrease or increase in weight ≤ 5% were defined as patients with stable weight.”
Comment 4:
(2) What are the cases of malnutrition in cancer? Is the intake reduced, or the energy usage increased bc of neoplasia?
Answer 4:
This depends on the mechanism of disease, as we discussed and tried to trace – head neck cancer patients have a locoregional impairment, that affects swallowing etc. First, these patients suffer from the effect of the cancer, later surgery and radiotherapy lead to impairment (e. g. loss of tissue through resection, xerostomia caused by radiation). High grade b cell lymphoma patients show a more systemic inflammation (reflected by the higher CRP at baseline before starting therapy or by B-symptoms which are usually triggered by a pro-inflammatory environment due to the lymphoma).
We tried to explain this patterns in the introduction:
“Currently, the field underestimates the influence of the distinct pathophysiological mechanisms inherent to different cancer entities. For example, high-grade B-cell lympho-ma (HGBCL) represents a systemic disease largely driven by inflammation, with immu-nochemotherapy being the standard of care [20,21]. In contrast, head and neck cancer (HNC) serves as a prime example of a malignancy where both the tumor itself and local treatments (e.g., surgery, radiotherapy) can directly impair nutritional intake due to local anatomical and functional disruption [22].” P2 l 60ff
Comment 5:
(3) Instead of referencing the guidelines of ESPEN and ACS, it is possible to show them as table in appendix?
Answer 5:
No, it is not, as the ESPEN guideline is a publication comprising 16 pages and the ACS 33 pages.
Comment 6:
(4) Line 55. Only in breast cancer? Could you please confirm?
Answer 6:
We wrote “predominantly “breast cancer. Actually this is true, because aromatase inhibitors lead to weight increase, which itself is a risk factor for cardiovascular events and breast cancer patients are often very compliant and adhere to follow-up and are motivated to participate in studies…. Other studies are scarce.
Comment 7:
(5) Line 59. Regarding "HGBCL) represents a systemic disease largely driven by inflammation". Please explain in detail the meaning of "inflammation" in your context. HGBCL is an hematological neoplasia within the spectrum of diffuse large b-cell lymphoma (intermediate with Burkitt/LBL).
Answer 7:
We are sorry, but we do not quite understand this point. Inflammation triggered by cancer as in lymphoma we have more a systemic disease, which is clinically reflected by B symptoms as well as for example the higher C reactive protein in the HGBCL group at disease onset compared to HNC patients.
Comment 8:
(6) Line 68. I understand using HNC, but why HGBCL? HGBCL is quite a rare disease and not representative of the most common hematological neoplasia.
Answer 8:
The aim of our paper is to demonstrate, that different cancer entities imply different mechanisms behind malnutrition. In order to detect difference, we chose two extremes (systemic disease vs. locoregional problem). Akute leukemia is even less common and more indolent hematological neoplasias have a lesser impact on nutrition.
Comment 9:
(7) Please introduce clinicopathological characteristics of both HNC and HGBCL in the Introduction sections.
Answer 9:
Clinicalpathological features relevant for this article (different mechanisms of disease) have been introduced. Further information on subentities do not offer additional information that is important to understand the article. We think going in more detail may confuse the reader from what is the aim of our study.
Comment 10:
(8) Line 86. Please add the statement of Helsinki for human experimentation.
Answer 10:
According to the journals author guidelines, this section is found under Institutional Review Board Statement p. 11 l295
“Institutional Review Board Statement: The study was conducted in accordance with the Declara-tion of Helsinki, and approved by the Institutional Review Board (or Ethics Committee) of University Medical Center Göttingen (protocol/approval number 12/6/24, approval date: 10.06.2024)).”
Comment 11:
(9) Line 94. Could you please add the mPGS in the Appendix as well?
Answer 11:
We added a short sentence how the score is calculated to the method section: "The score is calculated as following: albumin > 35 g/l and CrP < 10 mg/dl - 0 points; al-bumin > 35 g/l, CrP > 10 mg/dl – 1 point; albumin < 35 g/l, CrP > 10 mg/dl – 2 points [26,27]."
Comment 12:
(10) Line 103. How many cases had missing data? What percentage of the data was missing?
Answer 12:
We added the number of cases with weight per time-point to Table 1 so that the reader is also able to see how many absolute data points were available.
Comment 13:
(11) Line 110. Why not using COX as well for Aweight predictive value?
Answer 13:
A ROC curve demonstrates nicely sensitivity and specifity. Hazard ratios (Maental Cox) for patients with stable weight/weight gain compared to patients with weight loss at T2 also reflect this, please refer to Figure 2.
Comment 14:
(12) Line 113. I understand that "trend" is not significant.
Answer 14:
Yes, we defined this in the method section, it is a trend towards a development.
Comment 15:
(13) Line 121-122. Please describe HGBCL treatment (high-dose chemotherapy).
Answer 15:
We described treatment, please refer to p4 l 129ff
“HGBCL patients received systemic therapy according to national guidelines [21]. Within the HGBCL cohort, 11/97 received Chimeric-antigen-receptor-T cell therapy and 19/97 underwent high-dose chemotherapy followed by autologous stem cell transplantation after failure of first-line treatment.”
High dose treatment is depending on the sub-entity of HGBCL, often the BEAM protocol is used.
Comment 16:
(14) Figure 1 is in 3D. I would make it as 2D as the third dimension is not giving any additional informtion.(Of note, this comment is just a matter of style).
Answer 16:
Done.
Comment 17:
(15) In Figure 1, should time zero (at diagnosis) be added as well?
Answer 17:
This figure shows the number of patients, that experienced a weight change! This means, t0 is our baseline value and we see patients with stable weight/weight gain/weight loss at t1-t4 compared to baseline at t0.
Comment 18:
(16) In Table 1. I am not sure of the meaning of variable "intervention T0 and T1".
Answer 18:
We changed it to “nutritional intervention”
Comment 19:
(17) Figure 2 shows OS at differen times. As I understandn, T2 is at 6 months. I may be easier to remember "T 6months" instead of "T2".
Answer 19:
We defined the different time points several times in the text and chose to keep our nomenclature.
Comment 20:
(18) OS analysis was only done in HGBCL cases. Not having HNC data is a weak point.
Answer 20:
We know. We added this point to the limitation section.
Comment 21:
(19) Figure 3 shows the ROC analysis. However, difference of weight is also shown in Figure 2 with the K-M and log rank analysis, isn't it? Is not ROC analysis redundant?
Answer 21:
It is another statistical approach. Kaplan Meier only shows the difference between groups. ROC analysis answers the question, whether deltaweight is a sufficient biomarker and how good sensitivity and specifity of this marker is. This adds further information.
Comment 22:
(20) Line 168. C-reactive protein is a marker of inflammation. However, is albumin as well? Please confirm.
Answer 22:
Albumin is reduced in inflammation, as it is not an acute phase protein. Therefore, it is a part of the modified Glasgow Prognostic Score reflecting changes in metabolism.
Reviewer 4 Report
Comments and Suggestions for Authors
The authors have tackled an interesting topic in their work.
The Introduction provides a good introduction to the topic. The objective at the end of the Introduction is well-defined.
Materials and Methods:
I wondered what the recruitment process was like. How many patients were excluded and for what reason? Was there any other exclusion criterion besides the unavailability of body weight measurements? How many patients dropped out, and at what stage? Was it only due to the lack of these body weight measurements? A flow diagram would be helpful – one that would explain everything, so I kindly request the authors to provide such a diagram.
How many patients had more than three measurements (from the text, it appears that the maximum number of measurements was T1-T4)? My point is that the authors did not provide this information in the table, which could significantly affect the results. Since the inclusion criterion was a minimum of three measurements, did the majority have three or four? So, the number of patients at each time point would be necessary to include.
Similarly, the clinical description is relatively sparse. What were the stages of progression of the subjects? The description of the study group suggests that the treatment was varied: Either only lesion removal, or surgery + adjuvant chemotherapy/radiotherapy, or only radiotherapy/chemotherapy?
We know even less about HGBCL.
I believe that the stage of progression critically influences the study results.
In Table 1, I don't understand why the researchers calculated p when assessing differences for all groups, i.e., NHC+HGBCL; HNC; HGBCL (?).
At least that's what the table suggests. The results are presented as percentages (why combine them for these two groups?).
This is unclear to me.
In Figure 1, I feel like the results are repeated from the table. I believe 1a doesn't add anything. Two diseases are combined and then separated. This requires consideration and a decision on a single method of presenting the results.
I leave this to the researchers. In describing the results, the authors present the results again in the text, which is unnecessary.
It will be difficult for the reader to rely on the data about overall survival, since we don't know the stage of cancer progression, which, especially given the relatively small sample size of this single-center study, can significantly impact the survival curve.
Figure 4 is fascinating. I have no reservations.
In Table 2, I again question why these groups are combined. If they were compared with a completely different group, it would make sense, but it's not very defensible—this analysis.
Similarly, CRP. Here again, I have a question about the clinical status of the included patients. Can we rely on CRP if we don't know the patients' condition at these times? Perhaps they had acute inflammatory diseases? Treatment also affects CRP, as does supplementation.
I want to draw researchers' attention to how this should be presented.
In turn, are albumin levels affected by chronic kidney disease and liver disease? Were they excluded initially?
This is unknown. This information should certainly be supplemented.
The results of basic tests, complete blood counts, renal and liver functions would be helpful. This will allow a better understanding of which patients were included, and then the results will be assessed.
The conclusions are consistent with the intended purpose.
And my final question: Where did the idea come from to combine these specific disease entities for analysis?
And one more thing. I have a question of whether the patients had diabetes, hypertension, or chronic diseases—this cancer disease + comorbid condition as a combination of multiple morbidities, could also significantly influence the results. Could the researchers please explain this?
Generally, the article needs a deep improvement.
Author Response
Comment 1:
The authors have tackled an interesting topic in their work.
The Introduction provides a good introduction to the topic. The objective at the end of the Introduction is well-defined.
Answer 1:
Thank you.
Comment 2:
Materials and Methods:
I wondered what the recruitment process was like. How many patients were excluded and for what reason? Was there any other exclusion criterion besides the unavailability of body weight measurements? How many patients dropped out, and at what stage? Was it only due to the lack of these body weight measurements? A flow diagram would be helpful – one that would explain everything, so I kindly request the authors to provide such a diagram.
Answer 2:
We added the diagram as Supplemtary Figure 1.
Comment 3:
How many patients had more than three measurements (from the text, it appears that the maximum number of measurements was T1-T4)? My point is that the authors did not provide this information in the table, which could significantly affect the results. Since the inclusion criterion was a minimum of three measurements, did the majority have three or four? So, the number of patients at each time point would be necessary to include.
Answer 3:
We added the absolute cases of patients with weight at the time points to Table 1.
Comment 4:
Similarly, the clinical description is relatively sparse. What were the stages of progression of the subjects? The description of the study group suggests that the treatment was varied: Either only lesion removal, or surgery + adjuvant chemotherapy/radiotherapy, or only radiotherapy/chemotherapy?
Answer 4:
Treatment is adapted to the size of the lesion and differs from lesion side, especially in an entity like head neck cancer. This is why we stated, which of the three primary (curative) treatment options were offered (+ applied) to the patients.
Comment 5:
We know even less about HGBCL.
I believe that the stage of progression critically influences the study results.
Answer 5:
Patients were treated according to the guidelines, in 2021/2022 all patients received R-CHOP + 2x R or in the case of cerebral HGBCL a MTX-based chemotherapy regim. In case of recurrent disease high dose chemotherapy or CAR-T-cell therapy was applied. While patient with a lower IPI have a lower risk for recurrent disease, therapy is the same and patient number was to small for subgroup analysis by IPI or Ann Arbor stage. Our cohort comprised 32 patients with cerebral high grade b cell lymphoma (20 primary and 12 secondary b cell lymphoma), of patients with HGBCL without cerebral lesions, 27 were classified as Ann Arbor Stage I or II and 38 were classified as Ann Arbor Stage III or IV.
Comment 6:
In Table 1, I don't understand why the researchers calculated p when assessing differences for all groups, i.e., NHC+HGBCL; HNC; HGBCL (?).
At least that's what the table suggests. The results are presented as percentages (why combine them for these two groups?).
This is unclear to me.
Answer 6:
We compared HNC and HGBCL to test for differences between groups. We added a short sentence to Table 1. (“HNC and HGBCL patients were compared to detect differences between cohorts.”).
Comment 7:
In Figure 1, I feel like the results are repeated from the table. I believe 1a doesn't add anything. Two diseases are combined and then separated. This requires consideration and a decision on a single method of presenting the results. I leave this to the researchers. In describing the results, the authors present the results again in the text, which is unnecessary.
Answer 7:
This method of presenting the data visualizes the data of Table 1. Instead of comparing single values the bars allow a quick overview.
Comment 8:
It will be difficult for the reader to rely on the data about overall survival, since we don't know the stage of cancer progression, which, especially given the relatively small sample size of this single-center study, can significantly impact the survival curve.
Answer 8:
It is well known, that malnutrition and weight loss is associated with lesser survival in cancer patients of different cancer entities. We cite the respective studies in the discussion section.
As high grade B cell lymphoma are all treated similarly, as they are a systemic disease (so no TNM or UICC classifications is applied here and outcome is similar), our data is reliable. However, we concede, that the reviewer would make an important point, if we were talking about a solid cancer.
Comment 9:
Figure 4 is fascinating. I have no reservations.
Answer 9:
Thanks.
Comment 10:
In Table 2, I again question why these groups are combined. If they were compared with a completely different group, it would make sense, but it's not very defensible—this analysis.
Answer 10:
We show overall data of our mixed group and then the data per subgroup.
Comment 11:
Similarly, CRP. Here again, I have a question about the clinical status of the included patients. Can we rely on CRP if we don't know the patients' condition at these times? Perhaps they had acute inflammatory diseases? Treatment also affects CRP, as does supplementation. I want to draw researchers' attention to how this should be presented.
Answer 11:
As CrP is used for prognostic scores as standard of care and prognostic scores like the mGPS are well validated for using this score. A certain variation (e. g. due to infection) is to be expected. However, initial, significant differences between HGBCL (CrP available for 87 patients) and HNC (CrP available for 42 cases) CrP were observed at the time of diagnosis and we believe, that case number is high enough to make the assumption, that the pro-inflammatory environment of the HGBCL is carrying the difference between the two entities (especially, because this difference vanishes after t0, at t1 all patients are receiving treatment!).
Comment 12:
In turn, are albumin levels affected by chronic kidney disease and liver disease? Were they excluded initially? This is unknown. This information should certainly be supplemented. The results of basic tests, complete blood counts, renal and liver functions would be helpful. This will allow a better understanding of which patients were included, and then the results will be assessed.
Answer 12:
No, we did not excluded these casse. All patients included got standard of care treatment, which implies a sufficient kidney and liver function. We chose not to add these blood parameters, as these do not offer additional value to the observations we made.
Comment 13:
The conclusions are consistent with the intended purpose.
And my final question: Where did the idea come from to combine these specific disease entities for analysis?
Answer 13:
The authors treat both patient groups on their wards and felt, that usual standard of care and follow-up (“1-Size fits all”) is not sufficient for an individual, structured follow-up for cancer survivors. Our ENT-specialist noted, that HNC remained rather skinny, while our hematologists were often kind of happy, that their patients often regained weight (now, after diving into weight gain, we are adapting follow-up to account for metabolic syndrome).
Comment 14:
And one more thing. I have a question of whether the patients had diabetes, hypertension, or chronic diseases—this cancer disease + comorbid condition as a combination of multiple morbidities, could also significantly influence the results. Could the researchers please explain this?
Answer 14:
We present real world data. All this additional data points would be interesting in a very much larger cohort, but still the principles we base our research on 1) weight changes are relevant, 2) albumin and CrP are used as surrogate markers for inflammation and prognosis are well documented by many previous works. As mean age of our patients is > 60 years we have to expect age-related comorbidities. Age distribution between our two cohorts was not significantly different, therefore we worked under the premise, that datasets are comparable. As we treat all patients similarly according to guidelines and all patients included received standard of care, we have to assume that they were fit for therapy. And usually the comorbidities you listed do not result in weight loss.
Generally, the article needs a deep improvement.
Round 2
Reviewer 2 Report
Comments and Suggestions for Authors
The manuscript is approved.